# Evaluation of an Assertiveness Training based on the Social Learning Theory for Occupational Health, Safety and Environment Practitioners

Francisco J. Cantero-Sánchez [ID], José M. León-Rubio [ID], Raquel Vázquez-Morejón and José M. León-Pérez *[ID]

Cármides Research Group, Department of Social Psychology, Universidad de Sevilla, 41018 Sevilla, Spain; fcantero@us.es (F.J.C.-S.); jmleon@us.es (J.M.L.-R.); vazquezraquel@us.es (R.V.-M.)
* Correspondence: leonperez@us.es; Tel.: +34-955420075

**Abstract:** Assertiveness is a fundamental type of behavior for the creation and maintenance of positive relationships at work and the facilitation of team functioning. Therefore, the promotion of assertiveness contributes to improving work performance, preventing harassment at work, facilitating the adoption of safe behaviors and making critical decisions in terms of occupational health and safety. However, few studies have evaluated the effectiveness of assertive training to train occupational health, safety and environment (OHSE) technicians to facilitate their work as agents of change in the attitudes and behaviors of other workers. Therefore, an assertive training was carried out to increase assertiveness and decrease social anxiety in this type of professional. The training effectiveness was evaluated following a pretest–posttest group design. The results from both a pilot study in a sample of 328 undergraduate students and a study in a sample of 155 OHSE technicians indicated that the training was effective in achieving both objectives. Moreover, Cohen's d statistics suggest that the effect size was intermediate. These results are discussed with respect to their role in contributing to occupational health safety and environment practices as well as to the organizations' sustainability.

**Keywords:** assertiveness training; social anxiety; occupational health and safety; sustainability; social skills

## 1. Introduction

Social skills and assertiveness training can contribute to one of the most stimulating challenges that the European Commission (EC) has to face on its way to achieving the United Nations Sustainable Development Goals for 2030: ensuring a socially just transition to a sustainable economy [1].

To meet this challenge, the EC prioritizes social investment, one of the uses of which is an investment in human capital under the slogan "prepare rather than repair"; in other words, to prevent the consequences of the new social risks arising from the post-industrial society, which can be summarized as a lack of income and services [2]. Indeed, precarious employment, long-term unemployment, and the impossibility of balancing work and family are sources of poverty and social exclusion [3–5], so it is necessary to act before this happens; before people become vulnerable, it is necessary to invest in the development of their skills and the efficient use of them, so that they can cope with market transformations and have better opportunities to participate in society and the labor market [6].

In this regard, in May 2018, the Council of the European Union revised the 2006 Recommendation on key competences for lifelong learning, intending to "identify and define the key competences necessary for employability, personal fulfilment and health, active and responsible citizenship and social inclusion" [7] (p. 7) and proving a European reference framework to support efforts aimed at fostering the development of competences from a lifelong learning perspective.

This reference framework established eight key competences. Social skills and assertiveness training play a pivotal role in learning and developing such competences

because they require basic social skills such as interpersonal communication skills, team-work, problem-solving, or planning and organization [8]. In organizational contexts, these key competences are acquired through training. Indeed, the availability of a skilled workforce is crucial in the European social and economic model.

Employee training has innumerable advantages, both personal and social. On the one hand, it contributes to their well-being in terms of health, life satisfaction, and self-confidence, increasing their chances of remaining and progressing in employment, as well as better adjusting to changes in the labor market, if necessary [9]. On the other hand, training is an incentive itself, and employee motivation contributes to generating a good working environment [10] and close commitments to the objectives of the business organization for which they work [11], so that companies and society in general also benefit.

From this perspective, the aim is to strengthen key factors in order to confront new social risks and contribute to the sustainability of the welfare state. Among these factors, two stand out: (a) increasing productivity and efficiency in many areas through training, transfer of research results and the application of new technologies, and (b) creating responsible companies from a social and environmental perspective.

Based on the idea that assertive messages are more persuasive in promoting healthy and sustainable behaviors, this study aims to evaluate the effectiveness of assertiveness training for occupational health, safety and environment (OHSE) professionals, who are agents of change in the attitudes and behaviors of other workers [12]. In doing so, we follow Albert Bandura's Social Learning Theory [13,14], whose fundamental premises are that people learn new behaviors through the observation of socially relevant models and, with the practice of these new behaviors, they develop self-efficacy or confidence in their ability to achieve the intended results.

*OHSE Professionals: Linking Assertiveness and Sustainable Behaviors*

Assertiveness is defined as the skill to stand up for your interests in a positive way, by respecting others' rights and perspectives. Therefore, it is not surprising that being assertive has been related to job performance in organizational settings, mainly because assertiveness is a critical skill in the creation and maintenance of positive interpersonal relationships at work and effective team functioning [15–17].

In that sense, assertiveness training has an extensive background and history, particularly since the groundbreaking work of Wolpe [18]. Evidence suggests that assertiveness training, which is the core of any training in social skills, is still present nowadays due to its effectiveness in several settings, from therapy and the treatment of severe mental illness such as schizophrenia [19] to education and the improvement of social relations and the prevention of bullying [20,21].

In organizational contexts, assertiveness training is also a powerful tool to promote team functioning and prevent communication errors that may have fatal consequences for patients' or customers' safety in both healthcare and aviation industries [22–24]. Indeed, soft skills training has been associated with work performance rated by supervisors [25] and can contribute to the development of social corporate responsibility [26].

However, there is a lack of studies assessing assertiveness, communication, and social skills training in the field of occupational health, safety and environment (OHSE), a field of special relevance for organizations' sustainability given that: (1) OHSE practitioners perform a key role to convince other employees to adopt healthier and more sustainable behaviors. Indeed, their instructions are fundamental for making workers understand and align with organizational goals and avoid potential conflicts and ambiguity in the workforce when organizations try to promote environmental and sustainable outcomes [12,27]; and (2) assertiveness, from a communication point of view, plays a key role to persuade people and therefore is a fundamental social skill for leaders and managers, particularly when they need to convince, motivate and engage followers to do certain job tasks and adopt more sustainable behaviors [17,28,29]. Furthermore, a recent study has shown that positive language, when expressed assertively, is perceived as encouraging optimism

and self-efficacy, which facilitates the promotion of healthier, safer, and more sustainable behaviors [25,30].

Therefore, connecting the literature on assertiveness and social skills training at work with the evidence of assertive messages being more persuasive to promote healthy and sustainable behaviors, this study assesses the effectiveness of an assertiveness training for OHSE practitioners, who are agents of change in the attitudes and behaviors of other workers. Our results may have interesting implications for achieving the organization's sustainability goals. In assessing the effectiveness of our training, we test whether OHSE practitioners report being more assertive and experiencing lower levels of anxiety in social interactions after the training, which can be synthesized in the following hypotheses:

**Hypothesis 1 (H1).** *After the training, participants will report a significant improvement in their assertiveness scores.*

**Hypothesis 2 (H2).** *After the training, participants will report a significant improvement in their social anxiety scores.*

## 2. Materials and Methods

As this study reports the results of a training aimed at improving assertiveness, we first explain the rationale and content of the training; then, we describe the instruments (scales) used to evaluate the effectiveness of the training; and finally, we describe the procedure and participants enrolled in both the pilot study and the main study. We conducted a one-group pre–post training design in both cases (pilot and main studies).

### 2.1. Training Rationale and Content

We conducted an assertiveness training that consisted of 4 sessions of 2 h each and a one-hour follow-up session (9 h in total). We first tested our training in a pilot study with undergraduate students to check that all planned sessions and exercises were well understood, obtain some feedback about potential failures and develop our research protocol among trainers. Then, the training was implemented in our target sample: OHSE professionals. In conducting the training, we followed the principles of the Social Learning Theory [13,14], and therefore, each session comprised the following steps: (1) Instructions—In this first step, the instructions were communicated and explained to participants, and the trainer offered some examples and resolved doubts when necessary; (2) Modeling—After instructions, a role model successfully executing the techniques to be practiced was provided to participants. We created some videos to support our training activities and provided different role models (for example, for differentiating assertiveness from other communication styles: https://tv.us.es/estilos-de-comunicacion-interpersonal-entrevista-de-seleccionde-personal-i; accessed on 10 October 2021); (3) Practice—Participants performed role-play activities previously prepared by the authors. The main aim of this step was to apply in a practical way what was learned in the previous steps (instructions and modeling). Most of these activities were cooperative in nature and required being performed in small subgroups of three to four members; (4) Feedback—This step is crucial to let participants know if they are achieving the intended outcomes (i.e., improving their level of assertiveness). Feedback was given to the trainees or participants, focusing both on participants' strengths when performing role-playing activities and their weaknesses and elements that need to be improved; and (5) Generalization—This step was not carried out within the classroom, and it consisted of applying the content of the training and the learned techniques to the participants' job tasks.

In each session, participants learned specific topics and techniques [8,31–33]. In the first session, the pre-intervention data collection was carried out (although most participants filled out the questionnaires some days before when they received an email explaining the study and asking for their written informed consent). Then, trainers offered the theoretical rationale of the training and explained what assertiveness is and why it is an effective

communication style. Finally, following the steps mentioned above, participants conducted two activities: differentiating communication styles (i.e., assertiveness versus passive and aggressive styles) and applying such styles to several communication experiences that they have faced in their jobs; and communicating assertively by following three key steps to be assertive: listen actively and with empathy, express your needs and opinions and express your desired outcomes.

The second session focused on the practice of assertive communication and how to assertively refuse a request (assertive opposition or how to say 'no') through techniques such as empathic assertion (i.e., try to understand another person's feelings, needs, or interests) or the broken record technique (i.e., repeat the same message in different ways as often as necessary, in a calm relaxed manner). Then, the third session revolved around the ability to face and receive criticism and assertively request behavior change. In this third session, we included activities and techniques such as positive/negative inquiry (i.e., a way to deal with positive feedback or negative exchanges, respectively), formulating viable alternatives, or assertion of negative feelings (i.e., trying to control negative feelings towards another person or situation).

In the fourth session, participants practiced how to deal with complex and aggressive situations through techniques such as positive inner dialogues (i.e., to change negative predisposition into a more positive one that helps you confront a situation), or fogging (i.e., agreeing with some statements of the person who is being aggressive or is trying to manipulate you, but maintaining the integrity and your point of view). In this final session, the previous steps were also summarized and briefly reviewed.

Finally, there was a follow-up session one week later to check whether participants had efficiently used what they learned in class in their actual jobs. At the beginning of this session, post-intervention data were collected, and then participants discussed in the group potential limitations and barriers in extrapolating the in-class knowledge to their jobs, and they shared their own experiences about how to overcome such limitations and barriers.

### 2.2. Instruments and Measures

We measured the following variables both before (pre) and after (post) the training:

*Social anxiety* or level of discomfort and distress in social interactions was measured with the Assertion Inventory [34]. This inventory is a 40-item self-report scale that permits participants to report the degree of discomfort that they experience in the social situations described in each item (e.g., "Turn down a request for a meeting or date"; "Ask for constructive criticism"). Responses follow a Likert scale ranging from 1 (none) to 5 (very much). The inventory offers a total score ranging from 40 to 200, where higher scores represent a higher degree of discomfort in social situations (i.e., higher social anxiety). The reliability of the scale according to the Cronbach's alpha ($\alpha$ ranging from 0.92 to 0.94) was adequate in both samples.

*Assertive response* was also measured with the Assertion Inventory [34] in which respondents state the probability that they would engage in an assertive behavior in the given social situations (e.g., "Discuss openly with the person his/her criticism of your behavior"; "Express an opinion that differs from that of the person"). In this case, the inventory follows a Likert-scale ranging from 1 (always do it) to 5 (never do it), and therefore, the total score ranges from 40 to 200, where higher scores represent a lower probability of displaying an assertive behavior in social situations (i.e., lower assertiveness or assertive response). The reliability of the scale according to the Cronbach's alpha ($\alpha$ ranging from 0.86 to 0.92) was adequate in both samples.

*Overall assertiveness.* We complemented the measure of assertiveness with the General Assertiveness Scale [35]. This self-reported scale consists of 20 items in a Likert scale ranging from 1 (very often) to 4 (almost never) where respondents note for each item how often they behave assertively (e.g., "When I need something, I ask for it directly and with frankness"; "If someone treats me with contempt or condescension, I defend myself frankly without resorting to aggressiveness"). The scale provides a total score (from 20 to 80), where

higher scores represent lower overall assertiveness. The reliability of the scale according to the Cronbach's alpha ($\alpha$ ranging from 0.76 to 0.83) was adequate in both samples.

All analyses were performed using IBM SPSS® version 26.

### 2.3. Participants and Procedure

The procedure for both the pilot and the main study was the same. Participation was voluntary and written informed consent was required according to the ethical principles of the Spanish Association of Psychology. In line with the European general data protection regulation (GDPR), participants' names and emails were associated with a code in a dataset. When their responses to pre- and post-training tests were matched with such code, the dataset containing their names and emails was deleted. In that sense, to encourage participation and maintain anonymity and confidentiality, no other personal data were asked except their sex.

Data were collected in 2019 in two convenience samples. Regarding the *pilot study*, participants were first-year students enrolled in the course 'Social Psychology' in the School of Psychology at the University of Seville. In total, 328 students agreed to participate (76% women). These students were grouped into 2 main groups, each one with a trainer (A and B), and then subdivided into several small groups ranging between 15 and 25 students for implementing the training, which was conducted as part of their in-class activities during three consecutive weeks. In this pilot study, the training was compressed into three sessions of two hours each (6 h in total). After removing invalid responses and matching pre and post training measures, we obtained 254 valid responses (149 completed the social anxiety and assertive responses scales, and 249 on the overall assertiveness scale).

The participants in the *main study* were 155 Occupational Health, Safety and Environment practitioners (OHSE) that received the training as part of a capacitation program offered by their professional association. Participants were divided into five groups ranging from 30 to 40 members each (50.4% women). In this study, there were three trainers (the same two from the previous study and another one: A, B, and C). The training was conducted during five consecutive weeks in an out-of-work schedule (evenings from 7.00 to 9.00 pm). After matching pre- and post-training measures, we obtained 143 valid responses (all scales completed).

## 3. Results

This section is divided to describe the results of both the pilot study and the main study. In both studies, we first checked the assumptions of normality and variance homogeneity among groups. Moreover, we tested for differences in main variables depending on participants' sex (0 = men vs. 1 = women) and trainer (A vs. B vs. C), and we calculated descriptive statistics and bivariate correlations between the main variables. Finally, we tested the hypotheses.

### 3.1. Results of the Pilot Study

We performed a Kolmogorov–Smirnov test, and the results revealed that our variables for social anxiety and assertive response were normally distributed. However, the scores in the variable of overall assertiveness were non-normally distributed (K-S = 0.07, $p < 0.01$). Regarding homogeneity, Levene's test revealed that our variables met the assumption of equal variance (all $p$ values higher than 0.05). In addition, one-factor ANOVA analyses showed significant differences in the pre-training measures by the trainer, but not by sex. In that sense, participants in trainer A's group reported higher scores of social anxiety (M = 111.43; SD = 18.34) than their colleagues in trainer B's group (M = 100.11; SD = 23.28; $F(1,147) = 6.93$, $p < 0.01$). Table 1 presents descriptive statistics and bivariate correlations of our main variables.

**Table 1.** Means, standard deviations, and bivariate correlations among the main variables of the study (*n* = 149 for social anxiety and assertive response; 259 for overall assertiveness).

| Variable | Mean | SD | 1 | 2 | 3 | 4 | 5 | 6 |
|---|---|---|---|---|---|---|---|---|
| 1. Sex [a] | 0.24 | - | - | | | | | |
| 2. Social anxiety pre | 102.77 | 22.68 | −0.11 | - | | | | |
| 3. Assertive response pre [b] | 108.61 | 18.95 | 0.01 | 0.65 ** | - | | | |
| 4. Overall assertiveness pre [b] | 37.17 | 6.46 | −0.09 | 0.58 ** | 0.52 ** | - | | |
| 5. Social anxiety post | 88.45 | 21.74 | −0.12 | 0.63 ** | 0.40 ** | 0.41 ** | - | |
| 6. Assertive response post [b] | 101.31 | 20.02 | −0.05 | 0.38 ** | 0.52 ** | 0.39 ** | 0.57 ** | - |
| 7. Overall assertiveness post [b] | 34.09 | 7.00 | −0.09 | 0.41 ** | 0.37 ** | 0.67 ** | 0.56 ** | 0.58 ** |

Note: [a] percentage of men (24%); [b] higher scores represent lower assertiveness; ** $p < 0.01$.

In evaluating these results all together, we opted for testing the hypotheses with both parametric (paired-samples t-tests) and non-parametric tests (Wilcoxon). Our results showed that participants significantly reduced their social anxiety (*M*dif = 14.32; *SD* = 19.18; 95% CI (11.22; 17.43); *t*(148) = 9.12, *p* < 0.01) and increased their assertive responses (*M*dif = 7.30; *SD* = 19.04; 95% CI (4.22; 10.38); *t*(148) = 4.68, *p* < 0.01). Moreover, we used an online calculator to estimate the effect size in repeated measures designs [36] by taking into consideration the correlation between measurement points [37]. The effect size estimate revealed a pooled *d* = 0.749 (95% CI = 0.499; 0.969) for social anxiety and a pooled *d* = 0.382 (95% CI = 0.164; 0.622) for assertive response. These results suggest that the training was effective and had a small to intermediate effect size.

Furthermore, Wilcoxon signed-rank tests revealed that overall assertiveness was increased after the training (Z = 8.42, *p* < 0.01; 163 negative ranks with an average of 120.33 vs. 54 positive ranks with an average of 74.81). In this case, we followed Fritz and colleagues' recommendations for calculating effect sizes for nonparametric data [38]. Thus, after dividing the z value (8.42) by the squared root of the number of observations over the two time points (149 × 2 = 298), we obtained an effect size estimate of *r* = 0.487. In other words, the training had an intermediate effect size.

The effect size is the amount of gain measured in standard deviations; however, for a one-group with repeated measures design, how people change from one to another threshold of functioning can be more informative of the training's practical significance. In that sense, we used the categories suggested by Gambrill and Richey [34]. Thus, participants with scores higher than 96 in social anxiety are considered to experience high anxiety in social situations compared to low social anxiety participants, who score equal to or lower than 96. Similarly, participants scoring higher than 104 in assertive response can be labeled as low assertive participants compared to high assertive participants, who score equal to or lower than 104. According to such cut-off scores, we can create four categories: assertive (low anxiety and high assertive response), anxious assertiveness (high anxiety and high assertive response), careless of social situations (low anxiety and low assertiveness), and unassertive (high anxiety and low assertiveness). As can be seen in Table 2, chi-square tests showed that there was a change in both social anxiety ($x^2$(1,149) = 35.24; *p* < 0.01) and assertive response ($x^2$(1,149) = 38.21; *p* < 0.01) scores after the training: 27.5% of the participants moved from the high anxiety to the low anxiety category, suggesting that the training had a positive effect on them. On the other hand, 2.7% of the participants moved on the other way, from low to high anxiety, suggesting the training had a negative effect on them. Most of the participants (69.8%) remained in the same category after training. Similarly, 24.2% moved from low to high assertiveness after training (vs. 3.7% from high to low, and 72% remained equal). Furthermore, the chi-square test revealed that participants changed in their assertiveness categories after the training ($x^2$(9,149) = 88.22; *p* < 0.01): 23.5% of the participants improved their assertiveness level compared to 4% that diminished their assertiveness level and 72.5% that maintained the same assertiveness level after the training (see Table 3).

**Table 2.** Contingency table on social anxiety and assertive response over time (*n* = 161).

| Variable | | Social Anxiety Post | | | Assertive Response Post | | |
|---|---|---|---|---|---|---|---|
| | | **Low** | **High** | **(Total)** | **Low** | **High** | **(Total)** |
| Social anxiety pre//Assertive response pre | Low | 56 (37.6%) | 4 (2.7%) | 60 (40.3%) | 56 (37.6%) | 35 (23.5%) | 91 (61.1%) |
| | High | 41 (27.5%) | 48 (32.2%) | 89 (59.7%) | 6 (4%) | 52 (34.9%) | 58 (38.9%) |
| | (Total) | 97 (65.1%) | 52 (34.9%) | 149 (100%) | 62 (41.6%) | 87 (58.4%) | 149 (100%) |

**Table 3.** Contingency tables on assertiveness categories according to the Assertion Inventory (*n* = 161) and the General Assertiveness Scale (*n* = 279).

| Assertion Inventory Categories | | Post-Training | | | | |
|---|---|---|---|---|---|---|
| | | **Assertive** | **Anxious** | **Careless** | **Unassertive** | **(Total)** |
| Pre-training | Assertive | 36 (24.2%) | 2 (1.3%) | 2 (1.3%) | 1 (0.7%) | 41 (27.5%) |
| | Anxious Assertiveness | 6 (4%) | 8 (5.4%) | 1 (0.7%) | 2 (1.3%) | 17 (11.4%) |
| | Careless | 6 (4%) | 0 (0%) | 12 (8.1%) | 1 (0.7%) | 19 (12.8%) |
| | Unassertive | 20 (13.4%) | 9 (6%) | 14 (9.4%) | 29 (19.5%) | 72 (48.3%) |
| | (Total) | 68 (45.6%) | 19 (12.8%) | 29 (19.5%) | 33 (22.1%) | 149 (100%) |
| **General Assertivenes Categories** | | **Post-training** | | | | |
| | | High | High-Moderate | Moderate | Low | (Total) |
| Pre-training | High Assertiveness | 4 (1.6%) | 0 (0%) | 0 (0%) | 0 (0%) | 4 (1.6%) |
| | High-Moderate Assertiveness | 16 (6.4%) | 73 (29.3%) | 12 (4.8%) | 0 (0%) | 101 (40.6%) |
| | Moderate Assertiveness | 4 (1.6%) | 50 (20.1%) | 79 (31.7%) | 4 (1.6%) | 137 (55%) |
| | Low Assertiveness | 0 (0%) | 0 (0%) | 4 (1.6%) | 3 (1.2%) | 7 (2.8%) |
| | (Total) | 24 (9.6%) | 123 (49.4%) | 95 (38.2%) | 7 (2.8%) | 249 (100%) |

In a similar vein, overall assertiveness scores can be translated into four categories [35]: 20–25 = high assertiveness; 25–35 = high-moderate assertiveness; 35–50 = moderate assertiveness; 50–80 = low assertiveness. Accordingly, our results indicate that participants changed in their assertiveness categories after the training ($x^2(9,249)$ = 142.78; $p < 0.01$): 36.8% of the participants moved to a higher level of assertiveness category, 6% moved to a lower category and 57.2% remained in the same category after training (see Table 3).

*3.2. Results of the Main Study*

First, we assessed the normality and variance homogeneity assumptions for conducting parametric tests. In this sample, the results of the Kolmogorov–Smirnov test indicated that the scores of social anxiety were non-normally distributed (K-S test = 0.11 for pre-training measure, $p < 0.001$; and K-S test = 0.08 for post-training measure, $p < 0.05$).

However, Levene's test revealed that our variables met the assumption of equal variance or homogeneity (all *p* values higher than 0.05). Taking these results together, we opted for testing the hypotheses with both parametric (paired-samples *t*-tests) and non-parametric tests (Wilcoxon signed-rank tests).

In addition, one-factor ANOVA analyses showed no significant differences in the pre-training measures neither by sex nor by the trainer. Means, standard deviations, and bivariate correlations among the main variables of the study are presented in Table 4.

**Table 4.** Means, standard deviations, and bivariate correlations among the main variables of the study (*n* = 143).

| Variable | Mean | SD | 1 | 2 | 3 | 4 | 5 | 6 |
|---|---|---|---|---|---|---|---|---|
| 1. Sex [a] | 0.496 | - | - | | | | | |
| 2. Social anxiety pre | 97.19 | 23.36 | −0.07 | - | | | | |
| 3. Assertive response pre [b] | 109.41 | 16.14 | −0.03 | 0.32 ** | - | | | |
| 4. Overall assertiveness pre [b] | 36.87 | 6.92 | −0.09 | 0.40 ** | 0.32 ** | - | | |
| 5. Social anxiety post | 82.50 | 20.09 | −0.09 | 0.60 ** | 0.30 ** | 0.21 * | - | |
| 6. Assertive response post [b] | 98.01 | 19.81 | −0.01 | 0.34 ** | 0.65 ** | 0.19 * | 0.55 ** | - |
| 7. Overall assertiveness post [b] | 32.60 | 6.15 | −0.16 | 0.35 ** | 0.25 ** | 0.58 ** | 0.40 ** | 0.33 ** |

Note: [a] percentage of men (49.6%); [b] higher scores represent lower assertiveness; * $p < 0.05$; ** $p < 0.01$.

Regarding the effectiveness, Wilcoxon signed-rank tests revealed that social anxiety was reduced after the training ($Z = 7.95$, $p < 0.01$; 111 negative ranks with an average of 72.99 vs. 23 positive ranks with an average of 41.01). In addition, results from paired-samples *t*-tests showed that participants significantly increased both their assertive responses ($M$dif = 11.41; SD = 15.31; 95% CI (8.87; 13.93); $t(1,142) = 8.91$, $p < 0.01$) and their overall assertiveness ($M$dif = 4.26; SD = 6.01; 95% CI (3.27; 5.26); $t(1,142) = 8.49$, $p < 0.01$) after the training. These results suggest that the training was effective (see Figures 1 and 2).

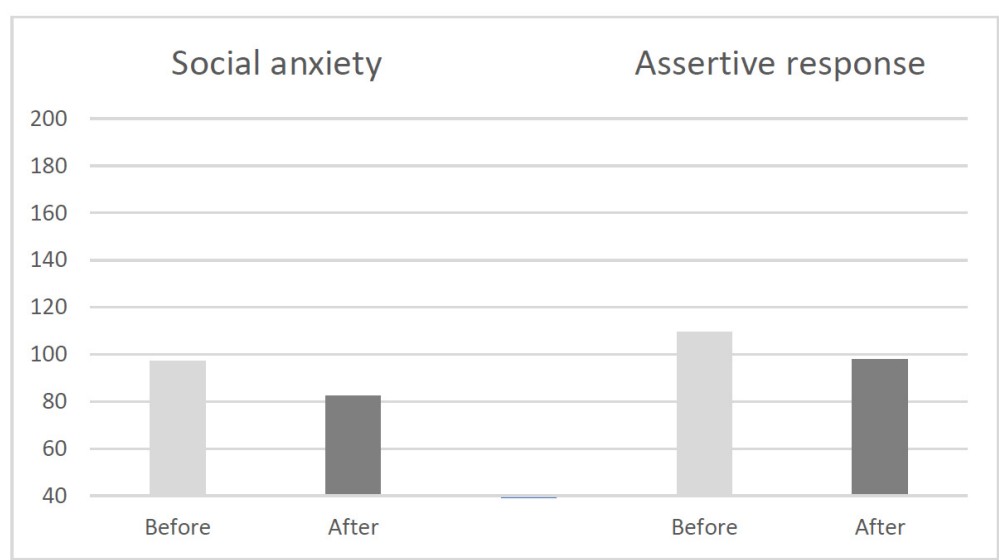

**Figure 1.** Scores in social anxiety and assertive response before and after the training.

We used an online calculator to estimate the effect size [36]. In the case of parametric tests, the effect size estimate in repeated measures design was used [37], which revealed a pooled $d = 0.754$ (95% CI = 0.602; 1.086) for assertive responses, and a pooled $d = 0.712$ (95% CI = 0.435; 0.912) for overall assertiveness. On the other hand, we followed Fritz and colleagues' recommendations for calculating effect sizes for nonparametric data [38]. Thus, after dividing the z value (7.95) by the squared root of the number of observations over the two time points ($143 \times 2 = 286$), we obtained an effect size estimate of $r = 0.47$. In other words, the training had a moderate or intermediate effect size.

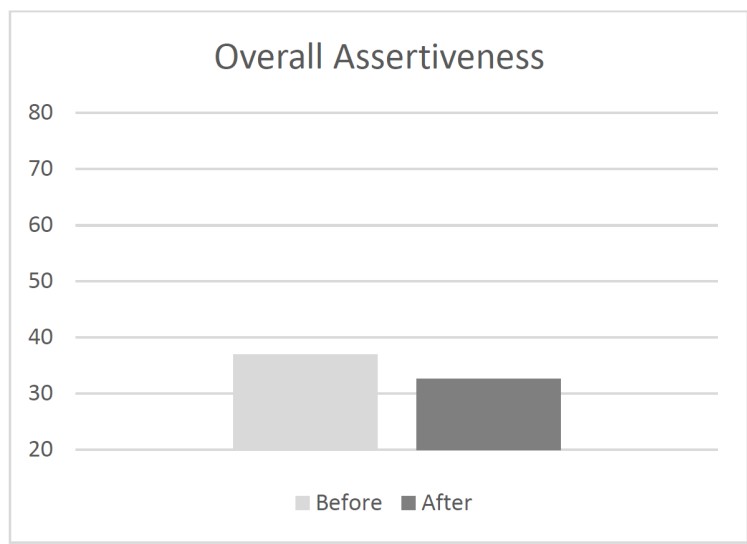

**Figure 2.** Scores in overall assertiveness before and after the training.

Finally, according to the categories suggested by the authors of the scales, we conducted chi-square tests and concluded that there was a change in social anxiety ($x^2$(1,142) = 19.77; $p < 0.01$) and assertive response ($x^2$(1,142) = 17.82; $p < 0.01$): 28% of the participants moved from the high anxiety to the low anxiety category, suggesting that the training had a positive effect on them (see Table 5). On the other hand, 4.2% of the participants moved on the other way, from low to high anxiety, suggesting the training had a negative effect on them. Most of the participants (67.8%) remained in the same category after training. Similarly, 34.3% moved from low to high assertiveness after training (vs. 4.2% from high to low, and 61.6% remained equal). Furthermore, the chi-square test revealed that participants changed in their assertiveness categories after the training ($x^2$(9,143) = 50.93; $p < 0.01$): 50.4% of the participants improved their assertiveness level compared to 6.3% that diminished their assertiveness level, and 57.9% that maintained the same assertiveness level after the training (see Table 6). In a similar vein, participants changed in their overall assertiveness categories after the training ($x^2$(9,143) = 43.16; $p < 0.01$): 37.1% of the participants moved to a higher level of assertiveness category, 4.9% moved to a lower category and 58% remained in the same category after training (see Table 6).

**Table 5.** Contingency table on social anxiety and assertive response over time (*n* = 143).

| Variable | | Social Anxiety Post | | | Assertive Response Post | | |
|---|---|---|---|---|---|---|---|
| | | Low | High | (Total) | Low | High | (Total) |
| Social anxiety pre//Assertive response pre | Low | 72 (50.3%) | 6 (4.2%) | 78 (54.5%) | 45 (31.5%) | 49 (34.3%) | 94 (65.7%) |
| | High | 40 (28%) | 25 (17.5%) | 65 (45.5%) | 6 (4.2%) | 43 (30.1%) | 49 (34.3%) |
| | (Total) | 112 (78.3%) | 31 (21.7%) | 143 (100%) | 51 (35.7%) | 92 (64.3%) | 143 (100%) |

**Table 6.** Contingency tables on assertiveness categories according to the Assertion Inventory and the General Assertiveness Scale (*n* = 143).

| Assertion Inventory Categories | | Post-Training | | | | |
|---|---|---|---|---|---|---|
| | | Assertive | Anxious | Careless | Unassertive | (Total) |
| Pre-training | Assertive | 30 (21%) | 0 (0%) | 1 (0.7%) | 1 (0.7%) | 32 (22.4%) |
| | Anxious Assertiveness | 10 (7%) | 3 (2.1%) | 4 (2.8%) | 0 (0%) | 17 (11.9%) |
| | Careless | 30 (21%) | 2 (1.4%) | 11 (7.7%) | 3 (2.1%) | 46 (32.2%) |
| | Unassertive | 13 (9.1%) | 4 (2.8%) | 13 (9.1%) | 18 (12.6%) | 48 (33.6%) |
| | (Total) | 83 (58%) | 9 (6.3%) | 29 (20.3%) | 22 (15.4%) | 143 (100%) |

| General Assertiveness Categories | | Post-training | | | | |
|---|---|---|---|---|---|---|
| | | High | High-Moderate | Moderate | Low | (Total) |
| Pre-training | High Assertiveness | 3 (2.1%) | 1 (0.7%) | 0 (0%) | 0 (0%) | 4 (2.8%) |
| | High-Moderate Assertiveness | 10 (7%) | 46 (32.2%) | 5 (3.5%) | 0 (0%) | 61 (42.7%) |
| | Moderate Assertiveness | 3 (2.1%) | 37 (25.9%) | 34 (23.8%) | 1 (0.7%) | 75 (52.4%) |
| | Low Assertiveness | 0 (0%) | 2 (1.4%) | 1 (0.7%) | 0 (0%) | 3 (2.1%) |
| | (Total) | 16 (11.2%) | 86 (60.1%) | 40 (28%) | 1 (0.7%) | 143 (100%) |

## 4. Discussion and Conclusions

Assertiveness training has been associated with job performance. Thus, promoting assertiveness in OHSE practitioners may contribute to achieving more environmental and sustainable organizational outcomes because they play an intermediate role between top managers in the company and bottom employees. Indeed, OHSE practitioners are in a privileged position to promote behavior change and transmit the organizational goals to bottom employees, who are the ultimate actors that should behave more environmentally and sustainably (of course with the support and the facilitation of the organization). In this process, assertiveness is crucial for the coworkers to successfully adopt the desired behaviors.

As expected, our results revealed that our training is effective, and OHSE practitioners improved the participants' assertiveness (and reduced their social anxiety) after the training. The effect size estimates indicate that the training has a moderate or intermediate effect. Beyond being significant from a statistical point of view, our results also showed that the training was significant from a practical point of view because one out of two participants (50.4%) improved their level of assertiveness. In other words, we can conclude that the training improved the social functioning perception of most participants when we used cutoff scores that classified participants into a certain assertiveness category based on their level of social anxiety or discomfort when facing social situations and the probability of giving an assertive response in such social situations.

The effectiveness of the training can be explained, among other factors, because we designed the training according to a strong theoretical framework and followed a participatory approach in the implementation that might have facilitated participants' commitment and engagement. In this sense, our results also confirm the utility of the Social Learning Theory [13,14], which is one of the most used theories to guide programs and training that develop social skills. Moreover, a participatory approach was implemented (e.g., role-playing techniques, group discussions), which facilitated the active involvement of participants in the planning and development of the training and therefore may have improved their motivation and intention to apply the training content to their daily

job [39]. Such a participatory approach has been successfully implemented to promote more sustainable communities and quality of life [40,41] and deserves further attention when implementing training programs aimed at promoting sustainable behaviors.

In addition, as trainers are crucial for achieving the desired outcomes and they may exert a great influence on the trainees [42], we emphasize the role of the trainers and encourage them to follow the same protocol and conduct the training identically. Furthermore, although we incorporated a third trainer in the main study, the pilot study provided useful information for developing a protocol on how to deliver the training. In that sense, our results did not reveal differences depending on the trainer who provided the training, and therefore, all groups successfully improved their levels of assertiveness after the training to a similar degree.

### 4.1. Limitations and Further Research

Although our study is rooted in a well-established theory and all the implementation factors mentioned above may have contributed to the success of training, there are also some limitations that need to be considered when trying to extrapolate our results to other samples or contexts. First, regarding the design, we did not include a control or comparison group, which prevents us from assuming that our results are exclusively due to our training as other factors may explain them. Although our results were similar in two different samples, future studies should incorporate a randomized-controlled trial design to offer more robust evidence. In doing so, further research should also include more measurement points after the training (follow-up measures) to evaluate the maintenance of the outcomes over time.

Second, following traditional training evaluation frameworks [43], we only assessed initial levels of the training effectiveness (skill acquisition). In the future, it would be of interest to evaluate both (1) the transfer of trained skills to the job, for example through third parties' reports on the assertiveness of the participants before and after the training, (2) the results of the training for the organization, for example through the evaluation of workers' sustainable behaviors before and after the training and (3) the return-on-investment (ROI) of this training. Indeed, the training seems to have affordable costs and interesting benefits: since just ten hours of training has proven to be effective for slightly more than half of the people trained to increase their level of assertiveness and decrease the social anxiety they experienced in difficult situations, it may be worthy to implement a similar training to promote more sustainable and healthy organizations. Further studies should offer more information on this issue and calculate the ROI of the training.

Third, cutoff scores to establish the assertiveness levels need to be replicated or adapted to other cultural contexts because there is a lack of evidence about their validity beyond the original studies conducted by the scales' authors.

Finally, as indicated in previous studies that have explored the role of assertiveness in leadership [28] and in the promotion of environmental behaviors [44,45], future research should bear in mind that assertiveness can reach a level that may lead to detrimental outcomes, at least under certain circumstances (i.e., curvilinear effects). Thus, boundary conditions that may determine positive or negative outcomes should be explored in the future.

### 4.2. Theoretical and Practical Implications

Despite these limitations, our results have interesting theoretical and practical implications. For example, it is a known fact that many workplace harassment situations in the hospitality and tourism industry, specifically sexual harassment, are not aborted in time because employees do not know how to block unwanted behaviors without offending customers [46], which could be achieved by training employees in assertiveness through a training program such as the one proposed in this study. Likewise, compliance with smoke-free workplace policies depends to a large extent on the assertiveness intentions of nonsmokers, which, in turn, seem to depend on the anxiety experienced by nonsmokers

in social situations or interactions with colleagues who smoke. Thus, interventions to reduce nonsmokers' social anxiety and train them in assertiveness could contribute to greater compliance with anti-smoking workplace rules [47,48], and the training procedure evaluated in this study has been shown to be effective in reducing the social anxiety of those trained.

Another issue of particular interest for organizational sustainability, more specifically for occupational safety, is how to decide on whether to continue or stop working when the working conditions have become unsafe because of unforeseen events or factors. Although there is some controversy about the role of workers' assertiveness to predict the decision to stop performing the work until conditions are safe again [49], assertiveness training may help to work in teams and make more effective group decisions.

In short, introducing assertiveness training and having an assertive group coaching procedure in the field of occupational health, safety and environment may constitute an effective measure not only for increasing assertiveness and reducing social anxiety in such professionals, but also for preventing workplace harassment, promoting smoke-free workplaces and making decisions that ensure the safety of workers, all of which contribute to the sustainability of work.

From a theoretical point of view, our results support the applicability of the Social Learning Theory's principles and contribute to a growing body of evidence in favor of assertiveness training to building proper relationships and ultimately create a good atmosphere at work.

In conclusion, this study offers a training that may contribute to achieving more environmentally friendly and sustainable organizational outcomes by enabling OHSE technicians to increase assertiveness and decrease social anxiety, which are essential behavioral repertoires for effectively influencing colleagues to adopt more responsible and sustainable behaviors.

**Author Contributions:** Conceptualization, J.M.L.-R. and J.M.L.-P.; methodology and data analysis, J.M.L.-P.; training and data collection, F.J.C.-S., J.M.L.-P. and R.V.-M.; writing—original draft preparation, F.J.C.-S. and J.M.L.-P.; writing—review and editing, J.M.L.-R. and R.V.-M. All authors have read and agreed to the published version of the manuscript.

**Funding:** This research received no external funding.

**Institutional Review Board Statement:** The study was conducted according to the guidelines of the Declaration of Helsinki. Ethical review and approval were waived for this study because less than minimal risk for participants was identified.

**Informed Consent Statement:** Informed consent was obtained from all subjects involved in the study.

**Data Availability Statement:** This study was conducted according to the guidelines of the Declaration of Helsinki and following the European regulations for personal data management. Ethical review and approval were waived for this study because data collection does not imply any risk for participants and does not use biological measures. Informed consent was obtained from all participants involved in the study. The data presented in this study are openly available on the website Open Science Framework: https://osf.io/npg3d/files (accessed on 10 October 2021).

**Conflicts of Interest:** The authors declare no conflict of interest.

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
