# Peer review of "Evaluation of an Assertiveness Training Based on the Social Learning Theory for Occupational Health, Safety and Environment Practitioners"

_sustainability, doi:10.3390/su132011504_

Round 1
Reviewer 1 Report
1.- The idea of the study seems interesting, different and even necessary.
2-There is no minimal and sufficient literature review regarding variables, literature section need to be improved . While establishing the hypotheses, the authors must give an extensive background .It needs a comprehensive review to justify the proposed hypotheses.
2- I recommend to add a separate literature review as a Title
4-The conclusions should be improved, including limitations and future lines of research. I Have seen some limitation and future lines under result , it should be under the conclusions , ex:"future research should bear in mind that 368 assertiveness can reached a level in which may lead to detrimental outcomes, at least un-369 der certain circumstances (i.e., curvilinear effects). Thus, boundary conditions that may 370 determine positive or negative outcomes should be explored in the future." . Also ,Theoretical and practical implications should be separated and more clear.
Others
5-The authors has to show how they solved the common method variance problem. Because, this issue has to be handled.
6- How the authors deal with the validity ?.
Author Response
Responses to Reviewer #1
1.- The idea of the study seems interesting, different and even necessary.
Authors (AU): Thank you very much for your appreciation.
2-There is no minimal and sufficient literature review regarding variables, literature section need to be improved . While establishing the hypotheses, the authors must give an extensive background .It needs a comprehensive review to justify the proposed hypotheses.
AU: Thank you for raising this issue. Now, we have provided a clearer rationale for our study, we have described better previous literature, and we finish the introduction with the formulation of the hypotheses (please, see lines 29-75).
2- I recommend to add a separate literature review as a Title
AU: Thank you for your comment. We have added a separate literature review title (“OHSE: linking assertiveness and sustainable behaviors”).
4-The conclusions should be improved, including limitations and future lines of research. I Have seen some limitation and future lines under result , it should be under the conclusions , ex: “future research should bear in mind that 368 assertiveness can reached a level in which may lead to detrimental outcomes, at least un-369 der certain circumstances (i.e., curvilinear effects). Thus, boundary conditions that may 370 determine positive or negative outcomes should be explored in the future." . Also ,Theoretical and practical implications should be separated and more clear.
AU: Thank you for your comment. We have added separate sections in the discussion: “Limitations and further research” and “theoretical and practical implications” (please, see the discussion section).
Others
5-The authors has to show how they solved the common method variance problem. Because, this issue has to be handled.
AU: We are aware that common-method variance (CMV) can introduce bias in cross-sectional research, mainly when the Independent Variable (IV) and the Dependent Variable are measured with the same method (e.g., a self-report in a survey) because variance can be attributable to the measurement method rather than to the constructs the measures are assumed to represent. However, as our design includes a manipulation (training) as IV, CMV bias hardly can represent a threat to our results.
6- How the authors deal with the validity ?
AU: We are sorry that we do not understand your question. As testing the psychometric properties and validity of our instruments is beyond the scope of this study (and we used scales that have been previously validated), could you please clarify which type of validity do you refer to? Construct validity was assessed with the scales’ internal consistency indicators (Cronbach’s alpha). Also, predictive validity was assessed by checking the increase/decrease in social skills after being exposed to the training.
Reviewer 2 Report
The article deals with an interesting topic. However, it has several shortcomings in terms of content.
The title of the article does not match the content of the article. It is not justified why the pilot study also appears in the article, when it follows from the abstract and the title that it is only a specific group (OHSE).
From the abstract it follows that it is an evaluation of assertive training, but on page 71 it is stated "Our training was intended to improve communication skills through being more assertive and experience lower levels of anxiety in social interactions." It is necessary to clarify what is the aim of the article and in this context to modify the abstract of the article.
The theoretical background is very weak. There is only the basic concept of assertiveness explained, but it is no longer explained social learning theory and there is also no link to sustainability. Also the conclusion is very weak.
On line 372 shows: As the training seems to have reduced costs since just ten hours of training....It is rather the subjective opinion of the authors, because they did not focus on this area in the research.
It is necessary to significantly rework the article to match the focus and level of the journal.
Author Response
Responses to Reviewer #2
The article deals with an interesting topic. However, it has several shortcomings in terms of content.
Authors (AU): Thank you very much for your valuable comments and suggestions.
The title of the article does not match the content of the article. It is not justified why the pilot study also appears in the article, when it follows from the abstract and the title that it is only a specific group (OHSE).
Authors (AU): Thank you for raising this issue. We have clarified that “We first tested our training in a pilot study with undergraduate students to check that all planned sessions and exercises were well understood, get some feedback about potential failures, and develop our research protocol among trainers. Then, the training was implemented in our target sample: OHSE professionals. In conducting the training (please, see lines 126-131). Indeed, we consider that a pilot study is always important to advance any potential failure of your design and, as we conducted the pilot study, we considered important to provide the results from such pilot study in line with the main international standards on ethical and open science.
From the abstract it follows that it is an evaluation of assertive training, but on page 71 it is stated "Our training was intended to improve communication skills through being more assertive and experience lower levels of anxiety in social interactions." It is necessary to clarify what is the aim of the article and in this context to modify the abstract of the article.
Authors (AU): Thank you for raising this issue. We have clarified the goal and contribution of our study: “Therefore, connecting the literature on assertiveness and social skills training at work with the evidence of assertive messages being more persuasive to promote healthy and sustainable behaviors, this study assesses the effectiveness of an assertiveness training for OHSE practitioners, who are agents of change in the attitudes and behaviors of other workers. Our results may have interesting implications for achieving organization’s sustainability goals. In assessing the effectiveness of our training, we test whether OHSE practitioners report being more assertive and experiencing lower levels of anxiety in social interactions after the training, which can be synthesized in the following hypotheses: H1: After the training, participants will report a significant improvement in their assertiveness scores. H2: After the training, participants will report a significant improvement in their social anxiety scores.” (please, see lines 110-116).
The theoretical background is very weak. There is only the basic concept of assertiveness explained, but it is no longer explained social learning theory and there is also no link to sustainability. Also the conclusion is very weak.
AU: We completely agree with this comment. Now, we have provided a clearer rationale for our study, we have improved the introduction (please, see lines 29-75) and the discussion, where we have integrated the conclusion at the end (please, see lines 476-480).
On line 372 shows: As the training seems to have reduced costs since just ten hours of training....It is rather the subjective opinion of the authors, because they did not focus on this area in the research.
AU: It is true that this comment is quite speculative and therefore it has been further elaborated in the “limitations and further research section” of the discussion (please, see lines 428-434): “and (3) the return-on-investment (ROI) of this training. Indeed, the training seems to have affordable costs and interesting benefits since just ten hours of training has proven to be effective for slightly more than half of the people trained to increase their level of assertiveness and decrease the social anxiety they experienced in difficult situations, it may be worthy to implement a similar training to promote more sustainable and healthy organizations. Further studies should offer more information on this issue an calculate the ROI of the training.”
It is necessary to significantly rework the article to match the focus and level of the journal.
AU: Thank you. In this version, following your suggestion, we offer a stronger theoretical framework focusing on how assertiveness can help to achieve sustainable goals, which match better the focus and level of the journal.
Reviewer 3 Report
The reviewed article concerns important issues related to evaluating assertiveness training important for practitioners of Occupational Health, Safety and Environment (OHSE). Therefore, it is of significant application importance, and in particular, the considerations presented in it may contribute to promoting the correct understanding of assertiveness, which may have an impact on building proper relationships and correct behaviors in the workplace, prevent and limit the negative phenomenon of mobbing and ultimately create a good atmosphere at work that helps to improve employee performance.
The following proposed modifications may increase the value of the article.
1. At the end of the Introduction section, it is worth emphasizing the key findings in the context of the main aim of the article. They are now defined too broadly as 'our results may have interesting implications for achieving organization's sustainability goals' (line 62-63). This is also a good place for reporting specific hypotheses being tested.
2. The first sentence of the Results section regarding the IBM SPSS version 26 software (in line 184) used has no relation to the next sentence concerning the division of the section into two parts. According to 'Instructions for Authors', the name and version of software used should be included in the Materials and Methods Section.
3. Considerations in the Discussion section should be carried out in the broadest possible context.
4. The Conclusions section is optional. It is worth considering whether the Authors should combine the Discussion and Conclusions sections because the latter now contains only one sentence that is difficult to define as a section.
5. According to the information presented in the article, 'the data presented in this study are openly available in the website Open Science Framework'. After logging in to this platform, a message appears that there is no access to this data and the following message 'You Need Permission' is displayed. The Authors should specify the restrictions on access to this data.
6. The scales of the vertical axes are wrongly selected in Figure 1. and Figure 2. (there is too much blank space).
7. Scientific papers are usually written in an impersonal style.
8. In line 23 should be: ‘with respect to their role’.
9. Confusing distribution of arguments and numbers in lines between 47 and 51.
10. The Authors shall correct some minor grammar mistakes; the verb ‘avoid’ requires a noun or gerund, e.g., avoid the idea that ... (line 50).
11. Punctuation needs to be corrected, e.g., in line 52.
12. Quite many repetitions concerning linking words, especially, furthermore, however and therefore, e.g., in lines 52, 54, 58.
13. The name of the first step is missing (line 81).
14. Wrong word order in line 84.
15. Any position referred to shall be numbered and placed within references (line 85).
Author Response
Responses to Reviewer #3
The reviewed article concerns important issues related to evaluating assertiveness training important for practitioners of Occupational Health, Safety and Environment (OHSE). Therefore, it is of significant application importance, and in particular, the considerations presented in it may contribute to promoting the correct understanding of assertiveness, which may have an impact on building proper relationships and correct behaviors in the workplace, prevent and limit the negative phenomenon of mobbing and ultimately create a good atmosphere at work that helps to improve employee performance.
Authors (AU): Thank you very much for your appreciation and stimulating comments.
The following proposed modifications may increase the value of the article.
1. At the end of the Introduction section, it is worth emphasizing the key findings in the context of the main aim of the article. They are now defined too broadly as 'our results may have interesting implications for achieving organization's sustainability goals' (line 62-63). This is also a good place for reporting specific hypotheses being tested.
AU: Thank you for raising this issue. Now, we have provided a clearer rationale for our study, we have described better previous literature, and we finish the introduction with the formulation of the hypotheses (please, see lines 29-75).
- The first sentence of the Results section regarding the IBM SPSS version 26 software (in line 184) used has no relation to the next sentence concerning the division of the section into two parts. According to 'Instructions for Authors', the name and version of software used should be included in the Materials and Methods Section.
AU: Thank you for your comment. Accordingly, we have moved this sentence to the methods section (instruments and materials).
Considerations in the Discussion section should be carried out in the broadest possible context.
AU: Following your suggestion, the discussion section has been improved. Also, we have added separate sections in the discussion: “Limitations and further research” and “theoretical and practical implications” (please, see the discussion section).
- The Conclusions section is optional. It is worth considering whether the Authors should combine the Discussion and Conclusions sections because the latter now contains only one sentence that is difficult to define as a section.
AU: Thank you for your suggestion. Accordingly, we have integrated the conclusion in the discussion section.
- According to the information presented in the article, 'the data presented in this study are openly available in the website Open Science Framework'. After logging in to this platform, a message appears that there is no access to this data and the following message 'You Need Permission' is displayed. The Authors should specify the restrictions on access to this data.
AU: Sorry for the error, it seems that all data were restricted. We are also all fully committed to progressively reach top open science standards, something fundamental in our field. Therefore, we have made all data publicly available.
- The scales of the vertical axes are wrongly selected in Figure 1. and Figure 2. (there is too much blank space).
AU: Thanks for pointing this out. We agree with your comment, as showing a smaller section of the response scale may facilitate the visualization of our results. However, our figures show the whole response scale to avoid “exaggerating” the level of change. In any case, if you consider them more appropriated, we will include them in the text replacing the previous ones.
- Scientific papers are usually written in an impersonal style.
AU: Thank you for raising this issue. There is some controversy about the style that scientific papers should follow. In that sense, we followed recent recommendations of the American Psychological Association (APA) to follow a first-person point of view to discuss research steps.
- In line 23 should be: ‘with respect to their role’.
AU: Corrected.
- Confusing distribution of arguments and numbers in lines between 47 and 51.
AU: Arguments have been clarified: “(1) OHSE practitioners perform a key role to convince other employees to adopt healthier and more sustainable behaviors. Indeed, their instructions are fundamental for making workers understand and align with organizational goals, and avoiding potential conflicts and ambiguity in the workforce when organizations try to promote environmental and sustainable outcomes [12,30]; and (2) assertiveness, from a communication point of view, plays a key role to persuade people and therefore is a fundamental social skill for leaders and managers, particularly when they need to convince, motivate, and engage followers to do certain job tasks and adopt more sustainable behaviors [20,31].”
- The Authors shall correct some minor grammar mistakes; the verb ‘avoid’ requires a noun or gerund, e.g., avoid the idea that ... (line 50).
AU: Corrected.
- Punctuation needs to be corrected, e.g., in line 52.
AU: Corrected.
- Quite many repetitions concerning linking words, especially, furthermore, however and therefore, e.g., in lines 52, 54, 58.
AU: Corrected.
- The name of the first step is missing (line 81).
AU: Corrected.
- Wrong word order in line 84.
AU: Corrected.
- Any position referred to shall be numbered and placed within references (line 85).
AU: Corrected.
Reviewer 4 Report
Dear authors,
I appreciate having the opportunity to review the manuscript entitled “Evaluation of an Assertiveness Training based on the Social Learning Theory for Occupational Health, Safety and Environment Practitioners” (sustainability-1368760).
This research evaluated the effectiveness of assertive training to train occupational health, safety, and environment technicians, to promote their work as agents of change in the attitudes and behaviors of co-workers. The assertive training was performed to enhance assertiveness and diminish social anxiety. Although the authors have made considerable efforts to develop this paper, however, I believe that the current version of manuscript should be improved through significant revision and re-writing. I want to provide some suggestions for the improvement of this paper as follows.
[1] Theories and hypotheses
- This paper did not provide the part of “Theory and Hypotheses. So, it is very difficult for me to be sure that the research has an enough level of theoretical value and contribution. I think that this is the critical flaw of this paper. Please provide the part in an elaborated way.
- Although this paper dealt with interesting phenomena, it did not provide adequate theoretical background and support for the development of its hypotheses. This is the critical limitation of this paper. Please clearly explain what its hypotheses are.
[2] Strengths and Limitations of the Study
- Although the authors have attempted to explain the contributions and implications of the paper, I think that the overall quality of the explanations is low. Please provide more elaborated explanations to demonstrate its theoretical and practical contributions.
I wish these comment may help you to improve your paper. Good luck.
Author Response
Response to
Reviewer #4
Dear authors,
I appreciate having the opportunity to review the manuscript entitled “Evaluation of an Assertiveness Training based on the Social Learning Theory for Occupational Health, Safety and Environment Practitioners” (sustainability-1368760).
This research evaluated the effectiveness of assertive training to train occupational health, safety, and environment technicians, to promote their work as agents of change in the attitudes and behaviors of co-workers. The assertive training was performed to enhance assertiveness and diminish social anxiety. Although the authors have made considerable efforts to develop this paper, however, I believe that the current version of manuscript should be improved through significant revision and re-writing. I want to provide some suggestions for the improvement of this paper as follows.
Authors (AU): Thank you very much for your appreciation and stimulating comments.
[1] Theories and hypotheses
- This paper did not provide the part of “Theory and Hypotheses. So, it is very difficult for me to be sure that the research has an enough level of theoretical value and contribution. I think that this is the critical flaw of this paper. Please provide the part in an elaborated way.
- Although this paper dealt with interesting phenomena, it did not provide adequate theoretical background and support for the development of its hypotheses. This is the critical limitation of this paper. Please clearly explain what its hypotheses are.
AU: Thank you for raising this issue. Now, we have provided a clearer rationale for our study, we have described better previous literature, and we finish the introduction with the formulation of the hypotheses (please, see lines 29-75).
[2] Strengths and Limitations of the Study
- Although the authors have attempted to explain the contributions and implications of the paper, I think that the overall quality of the explanations is low. Please provide more elaborated explanations to demonstrate its theoretical and practical contributions.
AU: Following your suggestion, the discussion section has been improved. Also, we have added separate sections in the discussion: “Limitations and further research” and “theoretical and practical implications” (please, see the discussion section).
I wish these comment may help you to improve your paper. Good luck.
Authors (AU): Thank you very much for your time and comments.
Reviewer 5 Report
Dear Author/s,
Re: Manuscript “Evaluation of an Assertiveness Training based on the Social Learning Theory for Occupational Health, Safety and Environment Practitioners”
Reviewer’s report:
The paper deals with an interesting topic such as the effectiveness of assertive training to train occupational health, safety, and environment (OHSE) technicians, which makes these professionals agents of change in the behaviors of other workers. The experiment has been carried out correctly and its limitations have been indicated at the end of the text. Likewise, the results, discussion and conclusions are also interesting. The modifications for the paper to be published would be the following:
- Justify the importance of studying and applying the study in Spanish groups.
- The date of the study is not provided.
- Nor are data offered on whether the sample size is sufficient for the study being carried out or its possible justifications.
Best regards
Author Response
Response to Reviewer #5
The paper deals with an interesting topic such as the effectiveness of assertive training to train occupational health, safety, and environment (OHSE) technicians, which makes these professionals agents of change in the behaviors of other workers. The experiment has been carried out correctly and its limitations have been indicated at the end of the text. Likewise, the results, discussion and conclusions are also interesting. The modifications for the paper to be published would be the following:
Authors (AU): Thank you very much for your time and comments.
- Justify the importance of studying and applying the study in Spanish groups.
AU: Thank you for your comment. However, we consider that is better to frame our paper with a more general or international focus. In that sense, we have opted for using the sustainable framework provided by the European Economic and Social Committee (please see the introduction).
- The date of the study is not provided.
AU: We have included your suggestion in the method section: “Data was collected in 2019 in two convenience samples.”
- Nor are data offered on whether the sample size is sufficient for the study being carried out or its possible justifications.
AU: Thank you for raising this issue. We did not offer data about the sample size because are convenience samples.
Round 2
Reviewer 2 Report
After the addition, the article is significantly better, but nevertheless there were minor shortcomings.
The discussion should be part of the results, as it currently has no explanatory power in such a state. Subchapters 4.1 and 4.2 should be part of the conclusion. The scientific article must have the conclusion, where should be summarizing the most important results of the analysis.
Author Response
Dear reviwer,
Thank you for your appreciation. Your valuable comments and suggestions to previous versions of the manuscript helped us to improve it.
We agree that the sections can be better structured. However, we consider that is better to mantain apart the results and the discussion sections. We opted for integrating the conclusion with the discussion and therefore we named this section as "Discussion and conclusions" (see line 376). We hope you welcome this change in response to your suggestion.
Thanks again for your time and valuable comments.
Sincerely,
The authors
Reviewer 4 Report
Dear authors,
Thank you for your efforts to revise your manuscript. I think that the paper is improved by the revision.
Author Response
Dear Reviewer,
Thank you for your comments and suggestions, which were useful to improve the manuscript.
Best Regards,
The authors